# Potassium homeostasis and signalling: from the whole plant to the subcellular level

Lars H. Wegner[1], Igor Pottosin[2] ⬤, Ingo Dreyer[3] ⬤ and Sergey Shabala[1,4] ⬤

[1]International Research Center for Environmental Membrane Biology, School of Agriculture and Bioengineering, Foshan University, Foshan, China; [2]Centro Universitario de Investigaciones Biomédicas, Universidad de Colima, Colima, México; [3]Electrical Signaling in Plants (ESP) Laboratory–Center of Bioinformatics, Simulation and Modeling (CBSM), Faculty of Engineering, Universidad de Talca, Talca, Chile; [4]School of Biological Science, University of Western Australia, Crawley, WA, Australia

## Review

**Keywords:**
long-distance transport; programmed cell death; autophagy; metabolic switch; stress response.

**Corresponding author:**
Sergey Shabala;
Email: Sergey.Shabala@uwa.edu.au

**Associate Editor:**
Dale Sanders

## Abstract

Potassium is an essential macronutrient required for plant growth and development. Over the recent decade, an important signalling role of $K^+$ has emerged. Here, we discuss some aspects of such signalling at the various levels of plant functional organisation. The topic covered include: (1) mechanisms of long-distant $K^+$ transport in the xylem and phloem and the molecular identity and regulation of $K^+$ loading and unloading into plant vasculature; (2) essentiality and physiological roles of $K^+$ cycling between shoots and roots; (3) plant sensing and signalling of low $K^+$; (4) maintenance of $K^+$ homeostasis at the cellular level; (5) stress-induced modulation of cytosolic $K^+$ as a signal in plant adaptive responses to hostile environment; (6) stress-specific $K^+$ "signatures" and mechanisms of their decoding by regulation of purine metabolism and $H^+$-ATPase activity; (7) cytosolic $K^+$ loss as a metabolic switch and a regulator of autophagy; and (8) vacuolar $K^+$ transport and sensing.

## 1. Introduction

Potassium is the eighth most abundant element in the continental crust (2.1%) and also in seawater (Benito et al., 2014). Potassium is indispensable for all known living organisms. Despite its high abundance, however, the $K^+$ food chain is not trivial. Animal organisms are not able to cover their K requirement from inorganic matter. Instead, they are dependent on plants that make this nutrient bioavailable in the first instance. This dependency may explain why potassium is likely the most investigated nutrient in plant science. In the model species *Arabidopsis thaliana*, 75 transporters may mediate $K^+$ movements across cellular membranes; 35 of these are $K^+$-selective (Very & Sentenac, 2003). Of these, high-affinity $K^+$ transporter AtHAK5 and the inward rectifier $K^+$ channel AKT1 are believed to be central to root $K^+$ uptake (Lhamo et al., 2021; Maierhofer et al., 2024; Rubio et al., 2020). The former transporter belongs to the HAK/KUP/KT $K^+$ transporter family that operates at external $K^+$ concentrations below 10 μM (Rubio et al., 2020), while the latter belongs to Shaker-like group of $K^+$ channels that mediate root $K^+$ acquisition at higher concentrations. The same is true for many other species such as rice (Nieves-Cordones et al., 2016), wheat (Zhang et al., 2019), soybeans (Chao et al., 2024), tomato (Rubio et al., 2014) and maize (Ma et al., 2020). Both types of transporters are regulated by complex pathways involving calcineurin B-like proteins (CBL) and CBL-interacting kinases (CIPK) and target of rapamycin (TOR) complexes that have been subjects of intensive studies (Cheong et al., 2007; Geiger et al., 2009; Li et al., 2023a,b) and are not discussed here. Similarly, for other aspects, we refer to several excellent review articles (Anschütz et al., 2014; Britto et al., 2021; Cakmak & Rengel, 2024; Ragel et al., 2019; Zhang et al., 2023; Zörb et al., 2014) and apologize that we cannot cite all due to space limitations. Instead, we have decided to focus here on an aspect that has not received too much attention to date but is receiving increasing attention in research: the role of $K^+$ as a signalling element.

## 2. K⁺ homeostasis at the whole plant level

Nutrient allocation within the plant is mediated by the two major vascular systems; the *xylem* provides a pathway for water and nutrient transport from the roots to the shoot, and the main role of *phloem* is in translocating sugars from photosynthetically active source tissues, mainly the leaves, to the roots and other below-ground organs as well as to seeds and fruits (Rogiers et al., 2017; van Bel & Hafke, 2005). The xylem is a microfluidic system formed by dead, interconnected tubes, whereas the phloem consists of living cells with flow occurring in the sieve elements. Via xylem and phloem, a circular transport of K⁺ is established (Figure 1).

Long-distance K⁺ transport, $J_{K+,i}$, in the compartment i (either xylem or phloem) can be simply described by the following equation:

$$J_{K+,i} = v_i * A_i * c_{K+,i} \tag{1}$$

with $v_i$, $A_i$ and $c_{K+,i}$ being the flow velocity, the cross-section of all xylem or phloem elements displaying flow and the xylem/phloem K⁺ concentration, respectively. An elegant way of assessing $v_i$ and $A_i$ non-invasively in intact plants is by the magnetic resonance imaging (MRI) technique. Using this technique, Windt et al. (2006) monitored xylem and phloem flow in poplar, tomato, tobacco and *Ricinus communis*. Phloem flow velocity ranged from 0.25 to 0.4 mm/s in all species during daytime; this corresponded to a volume flow rate ($v_i^* A_i$) of 0.1 to 0.2 mm³/s in the latter three species and somewhat higher values (~1 mm³/s) in poplar. The xylem flow velocity was higher by a factor of at least 10, and the volume flow rate ranged from 1 mm³/s in tomato to 17 mm³/s in poplar. Similar volume flow rates in the xylem and phloem during nighttime indicated the cycling of water between both compartments. For

intact transpiring plants, the xylem K⁺ concentration is directly accessible with the multifunctional xylem probe (Wegner & Zimmermann, 2002), combining a pressure probe with a K⁺ selective microelectrode. The location of the probe tip in a vessel is verified by recording pressure values below atmospheric or even vacuum level. In transpiring maize seedlings, xylem K⁺ concentrations of about 1 mM were measured, varying slightly, among other things, with K⁺ in the media. Root-to-shoot K⁺ export can also be quantified by combining xylem K⁺ recordings with gravimetrical measurements of water uptake (Wegner & Zimmermann, 2009). Measuring phloem K⁺ is also a technical challenge. By analysing the exudate from stylets of phloem-feeding aphids, Lohaus et al. (2000) obtained values of about 50 mM K⁺ for maize leaves, indicating that the phloem sap is ~50 times more concentrated than the xylem sap with respect to K⁺. If the xylem volume flow was exceeding that in the phloem by a factor of 50, this would imply a 100% K⁺ circulation between compartments. Unfortunately, we lack MRI data for maize, but this factor was 70 in tomatoes and lower in other species. Apparently, a very large part of xylem K⁺ ends up in the phloem and returns to the root.

Current estimates of the extent to which K⁺ is circulating are still based on less sophisticated techniques which rely on isolating xylem and phloem sap with invasive methods and measuring leaf and root K⁺ content (Jeschke & Pate, 1991; for possible technical limitations see Wegner, 2015). Values vary considerably depending on experimental conditions. For maize, Engels and Kirkby (2001), applying different temperature regimes to manipulate root and shoot growth, reported that between 34 and 93% of xylem K⁺ is recirculated via the phloem. Jeschke and Pate (1991) found values between 58.9 and 103 % for Lupin, *Ricinus communis* and barley at mild salinity. Touraine et al. (1988) investigated nitrate-fed soybean and observed that 68% of K⁺ moving upward in the xylem returned back to the root via the phloem. In plants reducing N in the shoot, nitrate is the main counterion for K⁺ in the xylem sap, whereas in the phloem it is frequently malate (so-called Ben Zioni-Lips-model; Ben Zioni et al., 1971). The extent to which K⁺ is recirculated depends on various factors including K⁺ shoot demand and the K⁺ nutritional status of the plant, suggesting that it serves as a shoot-to-root signal particularly under conditions of K⁺ deficiency.

The most compelling evidence in favour of this hypothesis comes from split-root experiments (Drew & Saker, 1984). K⁺ export from a single seminal barley root supplied with full nutrient solution was monitored using ⁴²K⁺ or ⁸⁶Rb⁺ as a tracer for K⁺, while the rest of the root system that was placed in a separate container was either fed with the same solution or deprived of K⁺. The latter treatment leading to K⁺ deficiency in the shoot strongly stimulated K⁺ export from the single, well-fed root and significantly reduced phloem K⁺ concentration in that root (Drew et al., 1990) compared to a situation of even K⁺ supply to all roots. Evidence for K⁺ re-circulation via the phloem also comes from blocking phloem transport, for example, in *Arabidopsis* mutants not expressing the NaKR1 (sodium potassium root defective 1) protein, leading to a K⁺ accumulation in leaves (Tian et al., 2010).

In order to understand how K⁺ allocation in the plant is organized, potential sites of control need to be identified. These are (i) xylem loading, that is, K⁺ release from stellar cells into adjacent dead xylem vessels in the root, (ii) K⁺ unloading from the xylem in leaf tissues and (iii) uptake of K⁺ by sieve tubes in assimilating leaf tissues. All three sites are associated with membrane transport steps that have been studied in some depth.

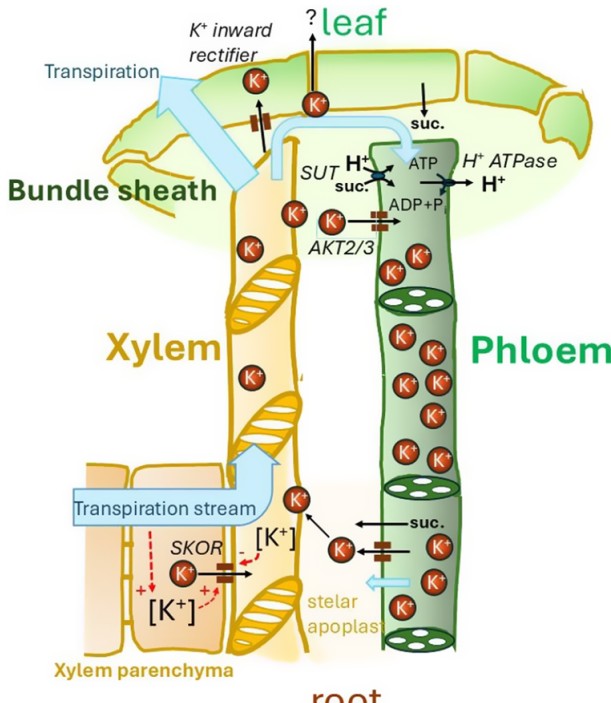

**Figure 1.** A schematic diagram showing K⁺ circulation via xylem and phloem in higher plants; suc., sucrose. For more details, see the text.

## 2.1. Xylem loading

Initial evidence for a key role of an outward rectifying $K^+$ channel in root xylem loading was already presented by Wegner and Raschke (1994) and Wegner and de Boer (1997). These authors performed patch clamp experiments on protoplasts isolated from barley xylem parenchyma. Later Gaymard et al. (1998) reported that a Shaker-type channel, denoted as a stellar $K^+$ outward rectifier (SKOR), served this function; in *Arabidopsis* SKOR knockout mutants the $K^+$ transport to the shoot was reduced by 50%. Particularly under conditions of nitrate deficiency, SKOR is tightly controlled by the activity of the nitrate transporter NRT1.5, co-expressed in xylem parenchyma cells (Drechsler et al., 2015), reminding us of the importance of a coordination of $K^+$ loading with anion transport. Interestingly, gating of SKOR is tightly regulated by its substrate concentration at either side of the membrane (Figure 1). $K^+$ transport capacity of SKOR increases with the cytosolic $K^+$ activity by enhancing the channel open probability (Liu et al., 2006). Hence, $K^+$ accumulation in xylem parenchyma cells, due to an increase in volume flow passing through these cells, would favour $K^+$ release into the xylem (so-called concentration-polarisation effect). Consistently, $K^+$ loading into the xylem was shown to strongly depend on transpirational water uptake by the roots (Wegner & Zimmermann, 2009). Additionally, SKOR gating was also shown to depend on the external $K^+$ concentration so that the shift of the equilibrium potential for $K^+$, $E_K$, was followed by almost equal shift of the voltage dependence of the channel open probability (Gaymard et al., 1998; Wegner & De Boer, 1997). Wegner and De Boer argued that $K^+$ channel activity and, hence, $K^+$ release into the xylem, will tend to decrease with the external $K^+$ concentration if the $K^+$ dependence of the cellular membrane potential (MP) of stellar cells is less pronounced than the shift in SKOR gating (which is likely, given the MP dependence on other conductances and the proton pump). This could be a way for $K^+$ recirculation via the phloem affecting, by means of its potential impact on the stelar apoplastic $K^+$ pool, SKOR gating. SKOR $K^+$ dependence has been extensively studied using site directed mutagenesis; the pore of the channel was shown to be also responsible for sensing the external $K^+$ concentration (Johansson et al., 2006).

Xylem $K^+$ concentration is also controlled along the way from the root to the leaves by cells bordering on the vessels. In Arabidopsis, AtHKT1.1 plays a key role in maintaining a favourable $K^+/Na^+$ ratio in plants suffering from salinity stress by mediating $Na^+$ resorption by these cells and stimulating $K^+$ release due to the related membrane depolarization (Hauser & Horie, 2010). Consistently, the $K^+/Na^+$ ratio was shown to be significantly lower in *athkt1.1* knockout mutants than in the wildtype.

## 2.2. Xylem unloading in the leaf

For $K^+$ unloading in the leaf, the bundle sheath cells (BSCs) enclosing the vascular tissue play an important role. Their effectiveness as a diffusion barrier towards the mesophyll is still under debate, particularly for C3 plants (Wigoda et al., 2014). In the C4 plant maize, $K^+$ is retrieved from the xylem sap via the BSCs, mainly in the smallest veins, as revealed by using $^{85}Rb^+$ as a tracer for $K^+$ (Keunecke et al., 2001). Inward-rectifying $K^+$ channels in *Arabidopsis* BSC protoplasts, which are likely to play a role in $K^+$ resorption from the xylem, differed from mesophyll $K^+$ channels with respect to their voltage dependence (Wigoda et al., 2017).

## 2.3. $K^+$ uptake and release by sieve tubes and the role of $K^+$ in the phloem

$K^+$ plays an important role in maintaining the source-to-sink translocation of sugars in the phloem, and, reciprocally, the phloem strongly contributes to $K^+$ homeostasis in the plant. Evidence for an involvement of $K^+$ in phloem sugar translocation came from the observation that even mild $K^+$ deficiency affected assimilate export from leaves (Amir & Reinhold, 1971). High phloem $K^+$ concentrations and the existence of a source-to-sink $K^+$ gradient (Vreugdenhil, 1985) prompted Lang (1983) to postulate a key role of $K^+$ in maintaining a phloem turgor gradient from source to sink, which is driving volume flow according to Münch's generally accepted 'Druckstrom' (pressure-flow) hypothesis. A $K^+$ gradient would particularly come into play when sucrose gradients fail to drive a sufficient mass flow. Lang argued that phloem $K^+$ gradients could be adjusted to optimize convective flow, which could thus even be uncoupled, to some extent, from processes of phloem *sucrose* loading and unloading at source and sink, respectively (see also van Bel & Hafke, 2005). A deeper insight into $K^+$ loading processes in the leaf came with the identification of the shaker-type $K^+$ channel AKT2/3, playing a key role in this process (Deeken et al., 2002). Interestingly, AKT2/3 can either be strongly voltage-dependent being activated by increasingly negative membrane potentials (but being inactive around the $K^+$ Nernst potential), or it can operate in a voltage-independent mode with $K^+$ channel activity being unaffected by the membrane potential (Dreyer et al., 2001). Based on these two modes of operation, Gajdanowicz et al. (2011) developed the $K^+$ battery concept (see also Dreyer et al., 2017 and Figure 1): The normally inward-rectifying channel can be switched by post-translational modifications, that is, phosphorylation, into the non-rectifying mode, which is mechanistically achieved by lowering the activation threshold of the channel (Michard et al., 2005). With these open channels, the transmembrane $K^+$ gradient, priorly established by $K^+$ accumulation in the sieve tube, then contributes to the electrical driving forces that energize other membrane transport processes, for example, sucrose loading into the sieve tube via a sucrose-$H^+$ symporter of the SUT type. The importance of the $K^+$ battery is particularly evident under energy-limiting growth conditions. When gating mode switching was disabled in transgenic *Arabidopsis*, the plants showed a retarded developmental phenotype at short days or reduced ambient $O_2$ supply causing local ATP deficiency (Gajdanowicz et al., 2011). The hypothesis of the $K^+$ battery was corroborated by a combination of experimental data and quantitative modelling. It should be noted that the mechanism relies on a sufficient $H^+$ buffer capacity of the sieve tubes and comes with a progressive alleviation of the $K^+$ gradient to maintain charge balance.

Phloem $K^+$ *unloading* at the sink, which received much less attention so far, can either be symplastic via plasmodesmata connecting the phloem with adjacent tissues or apoplastic by $K^+$ (and sucrose) release via the sieve tube plasma membrane. A shaker-type $K^+$ channel from *Vicia faba*, VFK1, is likely to be involved in the latter process; it was shown to be predominantly expressed in stems and in leaves becoming sinks by keeping them in the dark and depriving them of $CO_2$ (Ache et al., 2001).

To sum up, the main switches directing long-distance $K^+$ circulation in the plant may have been identified including some of the regulatory mechanisms, but we are still far from a truly systemic understanding of $K^+$ cycling between the xylem and phloem. $K^+$ allocation in the plant does not only rely on $K^+$ availability in the

soil and K$^+$ demand in source and sink tissues; it is also strongly affected by the convective mass flow in xylem and phloem (Wegner, 2015). Transpirational flow has a strong impact on K$^+$ delivery to the shoot both affecting the K$^+$ concentration gradient at the plasma membrane of root parenchyma cells *and* J$_{K+.x}$. On the other hand, mass flow in the phloem is dominated by assimilate synthesis in the source and demand in the sink. Moreover, K$^+$ transport in the xylem is linked to that of nitrate in plants reducing N in the shoot. Hence, we are confronted with a process integrating many aspects of plant life. Only a modelling approach taking account of these potentially antagonistic regulatory factors will allow us to unravel fine-tuning of K$^+$ allocation in the plant. This way, the molecular information on different K$^+$ transport processes will help us to understand phenomena of long-distance K$^+$ cycling known for several decades.

## 3. K$^+$ homeostasis at the cellular level

Modelling approaches have been proven already as rather helpful in filling gaps that are hard to address in wet-lab experiments. The above-mentioned concept of the K$^+$ battery in the phloem, for instance, was initially developed in computational simulations. Based on the experience from these dry-laboratory experiments pointed experiments with transgenic plants under energy-limiting conditions could be designed, in that case with low ambient O$_2$ to reduce ATP formation by respiration. The successful wet-laboratory experiment was thus the proof of concept of the model (Gajdanowicz et al., 2011; Sandmann et al., 2011). Recent advances in modelling have also provided new insights into K$^+$ homeostasis at the cellular level. The modelling of cellular steady-state conditions, initially thought to be 'trivial', revealed that an adjustable steady-state of the transmembrane K gradient can only be achieved if at least two differently energized K$^+$ transporter types are present in the membrane (Dreyer, 2021). If this condition is met, the transmembrane K$^+$ gradient can be adjusted by regulating the transporter activities. However, if it is not fulfilled, the regulation of transporter activity has no influence on the steady-state gradient (Dreyer, 2021; Dreyer et al., 2024). Remarkably, the gain of control over the transmembrane homeostatic condition has a metabolic cost. The steady state is inevitably accompanied by transmembrane H$^+$ and K$^+$ cycles driven by ATP-consuming proton pumping (Figure 2). Such nutrient cycling was reported experimentally, for instance, under salt stress (Munns et al., 2020; Shabala et al., 2020) and considered "futile" cycles (Britto & Kronzucker, 2006). The model analysis indicated, however, that these cycles are not as futile as they initially appear, but an essential part of the homeostatic process (Dreyer, 2021).

If the system is brought out of steady state by changing the potassium concentration on one side of the membrane, the transporter network (called homeostat) mediates a largely electroneutral H$^+$/K$^+$ antiport to restore the altered gradient to its original value. Such an H$^+$/K$^+$ exchange also takes place if there are no H$^+$/K$^+$ antiporters in the membrane via symporters and channels (Contador-Álvarez et al., 2025). The exchange of H$^+$ for K$^+$ upon a change in the K$^+$ gradient has been known since long (Blatt & Slayman, 1987; Conway & O'Malley, 1944; Maathuis & Sanders, 1994; Rodriguez-Navarro et al., 1986) and was also recently observed in guard cells and mesophyll cells from *Nicotiana benthamiana* or *Nicotiana tabacum* (Li et al., 2024). The homeostat concept now provides a detailed mechanistic explanation of these observations. Interestingly, H$^+$/K$^+$ antiport occurred in a coordinated manner

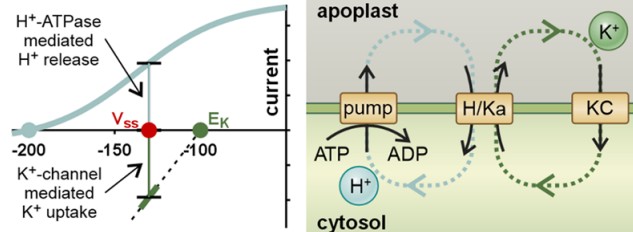

**Figure 2.** Homeostatic (steady state) condition of a transmembrane K$^+$ gradient. In this example, the membrane voltage in a steady state is $V_{ss} = -130$ mV. At this voltage, there is an efflux of H$^+$ released by the proton pump. The charge transport is compensated by a K$^+$ influx of the same magnitude via the K$^+$ channel (KC). Without further compensation, the system would not be stable in steady state but mediate an H$^+$/K$^+$ antiport, which would at least change the transmembrane potassium gradient. $E_K$ would become increasingly negative. The system would only be stable if the red and green dots overlap with the light blue dot at $-200$ mV. Stability at $V_{ss} = -130$ mV can be obtained with an additional H$^+$/K$^+$ antiporter. This transporter releases each accumulated K$^+$ ion and in the same process retrieves the released H$^+$ in an electroneutral manner. The system is in steady state, and $V_{ss}$ or $E_K$ do not change. However, there are ATP-consuming H$^+$ and K$^+$ cycles.

across both the plasma membrane and the vacuolar membrane supporting further the model prediction that there is an electro-chemical coupling to sequential membranes (Dreyer et al., 2023). Remarkably, besides organizing transmembrane K$^+$ transport the homeostat also generates a cytosolic pH signal that additionally depends on various cellular parameters (Contador-Álvarez et al., 2025). Thus, in addition to its role as an information carrier at the whole plant level, K$^+$ can also transmit messages at the cellular level.

The homeostat concept, which considers transporter networks instead of individual transporters, has so far explained well several observations, especially on K$^+$ homeostasis. However, it also provided new insights into the homeostasis of other ions and even hormones (Dreyer et al., 2022; Dreyer et al., 2024; Geisler & Dreyer, 2024; Li et al., 2024). As just mentioned before, homeostats appear also to constitute important parts of signalling processes in plants (Contador-Álvarez et al., 2025). Furthermore, the homeostat concept applies not only to the plasma membrane but also to endomembrane (Li et al., 2024). It could therefore contribute in the future to a better understanding of the role of K$^+$ at the subcellular level.

## 4. From the cellular to the subcellular level: stress-induced modulation of cytosolic K$^+$ as a signal in plant adaptive responses to hostile environment

In recent years, a significant bulk of evidence has been accumulated demonstrating that, in addition to a well-known role of K$^+$ as a nutrient, changes in intracellular K$^+$ concentrations may act as important developmental and adaptive signals (reviewed in Anschütz et al., 2014; Shabala, 2017; Wu et al., 2018), affecting key processes such as cell fate determination and autophagy. Stress-induced K$^+$ loss has been demonstrated in response to a broad range of abiotic and biotic stresses (reviewed in Shabala & Pottosin, 2014). These responses are often very fast (<1 min) and comparable with reported changes in cytosolic Ca$^{2+}$ signals that are traditionally considered being upstream of stress signalling (Dong et al., 2022). It was also shown that such stress-induced K$^+$ signalling is highly tissue-specific (Ahmed et al., 2021; Chakraborty et al., 2016; Shabala et al., 2016a), and that kinetics of stress-induced K$^+$ loss differ drastically between plant species, leading to a new concept of

the stress-induced K$^+$ flux 'signatures' (Rubio et al., 2020), similar to those reported for cytosolic Ca$^{2+}$. The physiological role of stress-induced modulation of cytosolic K$^+$ and the mechanisms by which these K$^+$ "signatures" are decoded are understood much less and require more discussion.

### 4.1. Potassium as a "metabolic switch"

This concept, first put forward by Demidchik (2014), implies that a drop in the cytosolic K$^+$ concentration, $[K^+]_{cyt}$, will inhibit energy-consuming anabolic reactions and allow plants to allocate more energy towards defence responses. Indeed, K$^+$ is essential for operation of numerous enzymes, by either providing proper ionic strength or catalysing their activity as a cofactor. For example, K$^+$ is critical for operation of the pyruvate kinase that catalyses the conversion of phosphoenolpyruvate (PEP) and adenosine diphosphate (ADP) to pyruvate and ATP in the glycolytic pathway, by enabling a proper electrostatic environment in the active site for reactants as well as for stabilization of the product (Cui & Tcherkez, 2021). Another example includes the succinyl-CoA thiokinase, an essential enzyme in tricarboxylic acid cycle catalysing the reversible conversion of succinyl-CoA to succinate, where K$^+$ facilitates magnesium binding and promotes the reaction in the forward direction (Lynn & Guynn, 1978). Potassium is also a key component of the ribosomes, being present at most crucial positions such as codon-decoding and peptidyl transferase domains (Rozov et al., 2019). This makes it essential for stabilization of mRNA binding and tRNA ligands, maintaining the correct frame position during elongation and reinforcing rRNA–protein interactions (Cui & Tcherkez, 2021). Most of these enzymes are designed to operate in a high mM K$^+$ concentration range. For example, $V_{max}$ value for starch synthase operation is 100 mM, 20 mM for glutathione synthase, 50 mM for pyruvate kinase, 90 mM for PEP carboxylase, etc (Evans & Sorger, 1966). At the same time, under stress conditions, the cytosolic K$^+$ level may drop several folds within minutes of stress exposure (e.g., down to 15 mM within 10 min of 50 mM NaCl treatment; Shabala et al., 2006). In lucerne plants, reduction in leaf K$^+$ content from 3.8 to 1.3 mg g$^{-1}$ DW resulted in a 3.5-fold reduction in Rubisco activity (Peoples & Koch, 1979), and starch synthase activity was reduced 6-fold by changing K$^+$ levels from 50 to 1 mM (Nitsos & Evans, 1969). Thus, a reduction in $[K^+]_{cyt}$ is expected to significantly reduce both plant enzymatic activity and protein synthesis avoiding the competition for energy between metabolic and defence responses. Consistent with this notion are reports that overexpressing pyruvate kinase PK21 (with strict K$^+$ dependency) has weakened soybean salt tolerance (Liu et al., 2024).

### 4.2. Potassium deprivation induces autophagy

Autophagy targets a wide variety of cytoplasmic cargo for degradation, including breaking down proteins into their constituent amino acids. Recently, Rangarajan et al. (2020) showed that K$^+$ starvation promoted autophagy in yeasts through core autophagy-related kinases and the PI3-kinase complex. In their work, transferring yeast cells from 10 to 1 mM K$^+$ media resulted in a 70% increase in the intensity of a Rosella signal, a fluorescent reporter of autophagy. Autophagy is also regulated by TORC1 (target of rapamycin complex 1). When $[K^+]_{cyt}$ is high, TORC1 remains active and inhibits autophagy via phosphorylation (Li et al., 2023b), and a drop in the $[K^+]_{cyt}$ deactivates TORC1 promoting assembly of autophagy-related proteins, with a 50-fold difference in the phosphorylation level of TORC-related genes reported for Arabidopsis plants after 2 h exposure to low-K$^+$ (10 μM) or high-K$^+$ (10 mM) liquid medium. TOR also interacts antagonistically with SnRK1 (sucrose nonfermenting 1 (SNF1)-related kinase 1), a heterotrimeric Ser/Thr protein kinase complex that is considered a key sensor of energy deficit and a central regulator of cellular energy homeostasis (Xiao et al., 2024). SnRK1 is an upstream regulator of TOR and can induce autophagy by directly interacting with autophagy-related proteins in the case of ATP shortage (Xiao et al., 2024).

### 4.3. Cytosolic K$^+$ levels can induce apoptosis

Apoptosis is characterized by a distinct series of morphological and biochemical changes that result in cell shrinkage, DNA breakdown, and, ultimately, cell death. Diverse external and internal stimuli trigger apoptosis, and enhanced K$^+$ efflux has been shown to be an essential mediator of not only early apoptotic cell shrinkage but also of downstream caspase activation and DNA fragmentation in many living systems (Peters & Chin, 2007; Remillard & Yuan, 2004). The failure to maintain high $[K^+]_{cyt}$ can also induce cell elimination via programmed cell death (PCD) by unblocking activities of caspase-like proteases and endonucleases in plants (Demidchik et al., 2010; Shabala, 2009).

### 4.4. Plant sensing and signalling of low K$^+$

In plant systems, the severity and specificity of stress is encoded by so-called cytosolic Ca$^{2+}$ "signatures" that are then decoded by a sophisticated Ca$^{2+}$ signalling network consisting of calcineurin B-like proteins (CBLs) and CBL-interacting kinases (CIPKs) that operate upstream of all key membrane transporters regulating plant ionic homeostasis (Dong et al., 2022; Li et al., 2023a). This is also true for low-K$^+$ stress, where CBL1/9 (calcineurin B-like 1/9) and CIPK23 (CBL-interacting protein kinase 23) interact with Ca$^{2+}$ signal triggered by low K$^+$, to cause reorchestration of metabolism (Rodenas & Vert, 2020). CBL1/9-CIPK23 complexes then phosphorylate and activate AKT1 (Behera et al., 2017) as well as regulate K$^+$ release from the vacuole (Li et al., 2023a), to maintain $[K^+]_{cyt}$ homeostasis. The latter authors showed that the K$^+$ status regulates the protein abundance and phosphorylation of the CBL-CIPK-channel modules at the posttranslational level. CIPK9/23 kinases were responsible for phosphorylation of CBL1/9/2/3 in plant response to low-K$^+$ stress and the HAB1/ABI1/ABI2/PP2CA phosphatases were responsible for CBL2/3-CIPK9 dephosphorylation upon K$^+$ repletion, acting as a negative regulator (Li et al., 2023a). The above responses to K$^+$ deprivation appear to be highly tissue-specific and rapid. Using Ca$^{2+}$ reporter protein YC3.6, Behera et al. (2017) reported an increase in cytosolic Ca$^{2+}$ starting just after 60 sec of K$^+$ deprivation and peaking at 140 sec. The response was most pronounced in the root elongation zone. Furthermore, Wang et al. (2021) identified a so-called post-meristematic K$^+$-sensing niche (KSN) where rapid K$^+$ decline and Ca$^{2+}$ signals coincided. The reported decline in K$^+$ was very rapid (observed within 60 sec and peaked at 2 min) and then triggered phosphorylation of the NADPH oxidases RBOHC, RBOHD and RBOHF, via CIF peptide-activated SGN3-LKS4/SGN1 receptor complex (Wang et al., 2021). It should be noted that low-K$^+$ signalling may also be intrinsically linked with Ca$^{2+}$ signalling, via voltage coupling. Indeed, a hyperpolarization of the membrane occurs when K$^+$ is removed which immediately leads to an increase in Ca$^{2+}$ influx through hyperpolarization-activated Ca$^{2+}$ channels

(Gelli & Blumwald, 1997; Grabov & Blatt, 1998; Lemtiri-Chlieh & Berkowitz, 2004) and, hence, a rise in the cytosolic $Ca^{2+}$.

Sensing of low-K environment also implies significant changes at the transcriptional level. Several $K^+$ transporters in Arabidopsis are transcriptionally induced by $K^+$-starvation, such as AtHAK5, AtKEA5, AtKUP3, AtCHX13 and AtCHX17 (Wang & Wu, 2010). Among the genes encoding the channels and transporters expressed in root outer cell layers, AtHAK5 appears to be the most highly regulated (Chérel et al., 2014). The relative abundance of OsHAK1 mRNA transcripts in rice seedlings was increased 2-fold 16 h after roots transfer to a $K^+$-free medium for 16 h (Bañuelos et al., 2002), and Nieves-Cordones et al. (2010) reported a 600-fold increase in *AtHAK5* transcript levels after 14 d of $K^+$ starvation. Low-K stress-induced increase in ROS levels seems to be required for these transcriptional changes (Shin & Schachtman, 2004) although the details remain scarce, as hundreds of other genes related to transcription, carbohydrate metabolism, hormonal signalling, protein phosphorylation, ROS metabolism and $Ca^{2+}$ signal generation and transduction, are also up-regulated in response to $K^+$-deficiency (Jin et al., 2024; Ruan et al., 2015; Schachtman & Shin, 2007; Shankar et al., 2013).

### 4.5. Decoding $K^+$ "signatures"

In the Arabidopsis genome, 26 CIPK and 10 CBL are present, providing a plethora of possibilities for decoding stress specificity. Such decoding machinery for $K^+$ "signatures" remains mostly unexplored. Here, two possible mechanisms are put forward and discussed.

**4.5.1. Decoding by regulation of purine metabolism.** One of the enzymes with a strong $K^+$ dependency is the adenosine kinase (ADK) that catalyses the phosphorylation of adenosine to $5'$-AMP using ATP as a source of phosphate. It was shown that binding of $K^+$ to $Asp^{310}$ is essential for a transition to a catalytically productive structure and removing auto-inhibition of ADK (de Oliveira et al., 2018). Thus, a drop in $[K^+]_{cyt}$ will reduce the amount of cAMP (3',5'-cyclic adenosine monophosphate). The latter, in turn, is a second messenger for a wide range of stress responses (Blanco et al., 2020), with demonstrated ability to regulate plant ionic homeostasis by controlling activity of various ion channels and transporters (Kaplan et al., 2007). One of them is SOS1 $H^+/Na^+$ exchanger that is considered as a critical component of the salt exclusion mechanism (Zhao et al., 2020). It was shown recently that SOS1 contains a cyclic nucleotide-binding domain (CNBD; Zhang et al., 2023) between residues 732 and 883. Moreover, mutations at G777 and G784 sides affected folding and stability of CNBD thus significantly decreasing transport activity of SOS1 (Wang et al., 2023). These findings suggest a possible causal link between salt-induced reduction in cytosolic $K^+$ content, ADK-mediated cAMP production and SOS1 activation. Also, in response to stress, small amounts of ATP (nanomolar concentrations; Xiao et al., 2024) are released to apoplastic space to be sensed by purinoreceptors that activate secondary signalling cascades (Kim et al., 2023).

**4.5.2. Decoding by regulating $H^+$-ATPase activity.** Stress-induced depolarization of the membrane potential (MP) is arguably one of the fastest events detected at the cellular level in response to a broad range of abiotic stresses (reviewed in Shabala et al., 2016). This is also true in response to $K^+$ deficiency, where the plasma membrane (PM) is hyperpolarized within a few minutes (Maathuis & Sanders, 1993; Nieves-Cordones et al., 2008), while

other signals such as ROS or ethylene start to operate after at least several hours (Behera et al., 2017). Equally fast are changes in MP caused by hypoxia or salinity (Bose et al., 2014, 2015) (Figure 3). As many $K^+$-permeable channels are voltage gated (Demidchik & Maathuis, 2007; Jegla et al., 2018), this comes with important implications for intracellular ion homeostasis. Recently, Maierhofer et al. (2024) have shown that AtHAK5 operates as a proton-driven $K^+$ transporter that is activated by membrane hyperpolarization, emphasising the essentiality of voltage gating in regulation of $K^+$ homeostasis in plant cells.

In plants, the critical contributor to MP maintenance is the $H^+$-ATPase pump. When the plasma membrane is depolarized under stress conditions (e.g., salinity stress), $H^+$-ATPase activity is rapidly upregulated (Chen et al., 2007; Shabala et al., 2016), to restore MP. Such activation also provides an additional driving force for secondary active ($H^+$-coupled) uptake of essential nutrients (i.e. $PO_4^-$, $NO_3^-$ or $K^+$) and exclusion of toxins (such as $Na^+$). $K^+$ can bind to the proton pump at a site involving $Asp^{617}$ in the cytoplasmic phosphorylation domain (Buch-Pedersen et al., 2006). Such binding can induce dephosphorylation of the $E_1P$ reaction cycle intermediate and, therefore, control the $H^+/ATP$ coupling ratio and uncouple ATP hydrolysis from transport. In other words, $K^+$-induced dephosphorylation serves as a negative regulator of the transmembrane electrochemical gradient. We believe that this mechanism is ideally suited for decoding stress-induced $K^+$ flux "signatures", as illustrated in an example later.

Earlier, we have shown that pea and wheat plants have different strategies of responding to salt. Pea plants were able to quickly restore their MP within minutes upon acute salt treatment (Bose et al., 2014) (Figure 3a), while the MP in wheat root epidermal cells remained depolarized for much longer period (Cuin et al., 2008) (Figure 3a). This difference was accompanied by a striking difference in the kinetics of the NaCl-induced $K^+$ loss from their roots (Figure 3b). In pea, this $K^+$ efflux was massive but transient while in wheat $K^+$ efflux was an order of magnitude lower and remained more or less constant over the first 10 min. Can these differences in $K^+$ flux "signatures" affect the operation of $H^+$-ATPase pumps and explain the differences in MP kinetics?

To answer this question, some model calculations were done. Assuming the diameter of the root cell being 50 μm, then the cell volume is $6.54 \times 10^{-14}$ $m^3$ (65.4 pL) and the cell surface area is $7.85 \times 10^{-9}$ $m^2$. Accepting a basal cytosolic $K^+$ concentration ($[K^+]_{cyt}$) as 100 mM, then the total amount of $K^+$ in one cell is 6.54 pmol under control (unstressed conditions). In pea plants, the total $K^+$ loss from root cells ranged from ~ 520 nmol $m^{-2}s^{-1}$ 1.5 min after stress onset to ~ -40 nmol $m^{-2}s^{-1}$ 20 min after (Figure 3b). The total $K^+$ loss (in nmol) from the cell can be calculated as

$$[K \ loss] = Flux \ (nmol \ m-2s-1) \times surface \ area \ (m2) \times time \ (sec) \tag{2}$$

Assuming that the efflux measured by non-invasive MIFE technique originates from the topmost (epidermal) cell layer; the cytosol representing 10% of the cell volume; no vacuolar buffering (e.g., no ability to replace lost cytosolic $K^+$ from the vacuolar $K^+$ pool); and feeding the flux data from Figure 3b into (Eq. 2), one can see that $[K^+]_{cyt}$ in pea will drop by about 2-fold within 2 min after stress onset and will be reduced to zero within 6 min (Figure 3c). At 5% vacuolar buffering capacity, a 50% decline in $[K^+]_{cyt}$ will occur by 6 min, and at 10% buffering – at 18 min (Figure 3d). Such rapid and drastic changes in $[K^+]_{cyt}$ may explain rapid activation of $H^+$-pumping (Bose et al., 2014) and

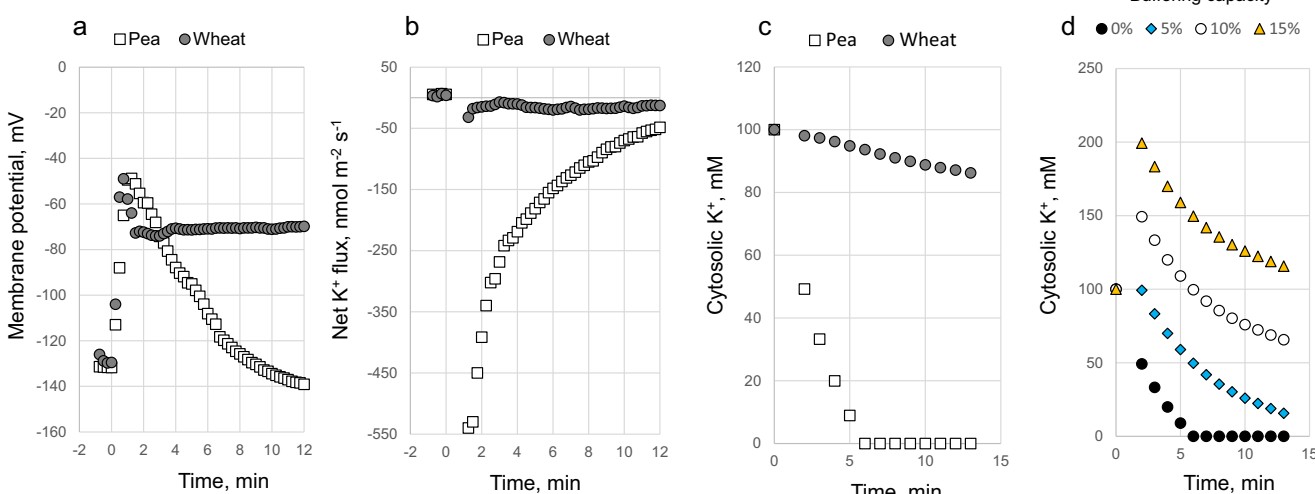

**Figure 3.** Difference in NaCl-induced K$^+$ flux 'signatures' explain the difference in the strategy of adapting to salt stress between pea and barley. (a) Transient changes in MP of epidermal root cells in wheat and pea in response to acute NaCl treatment (based on Cuin et al., 2008 and Bose et al., 2014, respectively). (b) Net K$^+$ fluxes measured from plant roots by non-invasive ion flux measuring technique. (c) Predicted changes in cytosolic K$^+$ concentration in pea and wheat root cytosol assuming no buffering from the vacuole. (d) Effect of vacuolar K$^+$ buffering on cytosolic K$^+$ kinetics in pea roots exposed to acute salinity treatment. Percentage values are the size of the vacuolar K$^+$ available to be used to compensate cytosolic K$^+$ loss.

repolarization of the MP (Figure 3a) following the removal of K$^+$-induced dephosphorylation of H$^+$-ATPase. In wheat, however, NaCl-induced reduction in $[K^+]_{cyt}$ was much slower, and even without vacuolar buffering within 15 min $[K^+]_{cyt}$ will drop only by 15 mM (Figure 1c). As a result, the H$^+$-ATPase remained dephosphorylated for much longer and failed to rapidly restore the MP.

As shown in the above-mentioned examples, the extent of H$^+$-ATPase dephosphorylation will be critically dependent not only on the magnitude of stress-induced K$^+$ loss but also on the plant's ability to replenish $[K^+]_{cyt}$ loss from the vacuolar pool which may be highly specific for different cell types/tissues, as well as plant species. Thus, modulation of the coupling between H$^+$-ATPase hydrolytic and transport activity by $[K^+]_{cyt}$ "signatures" may represent a decoding mechanism and explain the difference in species' strategies of handling salt stress.

The above modelling has been done without considering the fact that cells are connected by plasmodesmata, so cortical cells may also provide some additional buffering. However, giving the lack of the quantitative data on how rapid K$^+$ from the neighbouring cells may be transported to root epidermis, modelling of this process is nearly impossible. It should be also kept in mind that these (cortical) cells will be also exposed to NaCl stress and will undergo the same membrane depolarisation and, hence, K$^+$ loss. Thus, such additional "buffering capacity" is likely to play a relatively minor role in determining cytosolic K$^+$ levels in root epidermal cells.

## 5. Vacuolar K$^+$-permeable channels sense vacuolar K$^+$ content

The vacuole is the main intracellular K$^+$ store in plant cells and, except extreme K$^+$-starving conditions, the electrochemical K$^+$ gradient across the tonoplast is cytosol-directed. Thus, K$^+$ accumulation in the vacuole is mediated by an active K$^+$/H$^+$ antiport, whereas passive vacuolar K$^+$ release is channel-mediated (Li et al., 2024). The vacuolar K$^+$ concentration tends to be more variable as compared to cytosolic K$^+$. During stomata closure there is a

massive decrease of intracellular (presumably, mainly vacuolar) K$^+$ content, from 450 to 95 mM in *Commelina communis* (Leigh, 1997). For several plant species, collectively, upon stomatal closure the cytosolic K$^+$ concentration decreases from 150–247 mM to 55–93 mM, whereas the vacuolar one from 181–454 mM to 38–92 mM, (Jezek & Blatt, 2017). In open stomata after the midday, K$^+$ is substituted by 50% with sugars (Talbott & Zeiger, 1996). In salinized barley leaves K$^+$ dropped (substituted by Na$^+$) both in the vacuole and the cytosol; these changes were dramatic (about 5-fold) in epidermal cells, and less prominent in mesophyll cells, where cytosolic K$^+$ tends to be maintained at the expense of the vacuolar K$^+$ pool (Cuin et al., 2003), thus improving cytosolic K$^+$/Na$^+$ ratio, underlying a higher photosynthetic activity and salt tolerance (James et al., 2006). The refilling of the cytosol by vacuolar K$^+$ is most pronounced in the case of K$^+$ deficiency: the vacuolar K$^+$ activity can drop 10-fold, whereas the cytosolic K$^+$ activity remains almost constant (Walker et al., 1996).

There are three types of ion channels, which mediate passive K$^+$ fluxes across the tonoplast: (i) VK (K$^+$-selective) channel, mainly encoded by TPK1, (ii) slow vacuolar (SV) channels; non-selective cation channels encoded by TPC1 and (iii) fast vacuolar (FV) channels (also non-selective; with unknown molecular identity). Only VK/TPK channels, but not SV/TPC1 or FV, can discriminate between K$^+$ and Na$^+$ (Brüggemann et al., 1999a; Gobert et al., 2007; Pottosin et al., 2003; Pottosin & Dobrovinskaya, 2014; Ward & Schroeder, 1994). FV and SV channel activity and/or conductance is modulated by a variety of cytosolic and vacuolar factors, including Ca$^{2+}$ (and Mg$^{2+}$), voltage and polyamines. FV activity is suppressed by cytosolic Ca$^{2+}$, Mg$^{2+}$ and polyamines in the micromolar–submillimolar range (Brüggemann et al., 1998, 1999b; Dobrovinskaya et al., 1999; Tikhonova et al., 1997). In the virtual absence of these cations at both tonoplast sides and symmetric K$^+$ in a physiological range, it is activated at an increased voltage difference of either sign (Allen et al., 1998; Bonales-Alatorre et al., 2013; Pottosin et al., 2003; Tikhonova et al., 1997), with a minimum open probability at about resting tonoplast potential of −20 to −30 mV (Miller et al., 2001; Walker et al., 1996; Wang et al., 2015). Vacuolar

polycations at millimolar concentrations convert FV into an outward rectifier, suppressing the cytosol-directed $K^+$ flux (Tikhonova et al., 1997). The SV/TPC1 channel is activated at cytosol-positive voltage and its opening requires high (ten micromolar) cytosolic $Ca^{2+}$, whereas vacuolar $Ca^{2+}$ (and $Mg^{2+}$) locks the channel in the resting state, thus increasing the threshold for its voltage activation (reviewed in Pottosin & Dobrovinskaya, 2022). VK/TPK channels are mainly expressed in guard cells and to a lesser extent in root and mesophyll, yielding 10 to 200 VK copies per vacuole (Pottosin et al., 2003; Tang et al., 2020; Ward & Schroeder, 1994), whereas a typical vacuole contains thousands of active copies of SV/TPC1 and FV channels. Yet, due to the fact that VK/TPK channels are voltage-independent, insensitive to $Mg^{2+}$ and polyamines and only require elevated (>1 μM) cytosolic $Ca^{2+}$, which can activate the channel directly or via the CBL-CIPK pathway (Allen et al., 1998; Pottosin et al., 2003; Tang et al., 2020; Ward & Schroeder, 1994), it is conceivable that they can dominate $K^+$ flux from vacuole at moderate cytosolic $Ca^{2+}$ elevations. Strict negative control of SV/TPC1 channels by physiologically relevant factors can question its direct contribution to tonoplast $K^+$ and $Ca^{2+}$ fluxes. However, it has been shown that in vivo SV/TPC1 activation by a depolarization is transmitted in some way to the VK/TPK activation, mediating vacuolar $K^+$ release by the latter, as if the activity of these two tonoplast channels is functionally coupled (Jaslan et al., 2019). It is conceivable that such a transduction occurs in a direct mechanical way so that the movement of the voltage sensor of the SV/TPC1 channels favours the activation of associated VK/TPK ones; for such mechanism full activation (opening) of SV/TPC1 channels, which is only observed at rather high cytosolic $Ca^{2+}$, is not necessary so that the channel protein may operate in a non-canonical ion flux-independent way (Pottosin & Dobrovinskaya, 2022).

VK/TPK channels were proven to play key roles in vacuolar $K^+$ release upon ABA-induced stomatal closure (Gobert et al., 2007) and vacuolar $K^+$ remobilization upon $K^+$ starvation (Gu et al., 2024; Tang et al., 2020). VK/TPK activity is postulated to be predominant over the activity of non-selective SV/TPC1 and FV channels under salt stress, not only for the means of refilling the cytosolic $K^+$ pool but also for a proper $Na^+$ compartmentalization, avoiding a futile energy-consuming vacuolar $Na^+$ cycling (Shabala et al., 2020; Wu et al., 2018).

Vacuolar $K^+$ dynamics appears to be controlled by the vacuolar $K^+$ content. Firstly, upon the increase of the $K^+$ concentration in the growth medium, $K^+$ cellular (principally, vacuolar) content tends to approach the upper limit of 100–250 mM when external $K^+$ reaches 24–95 μM (species-dependent) and does not increase upon further increase of external $K^+$ to 1 mM (Leigh, 2001). On the other hand, under stomatal closure, induced by low and optimal ABA concentrations, the release of the vacuolar $K^+$ analogue $^{86}$Rb, albeit substantially slower at low ABA dosage, ceased when the same loss of vacuolar tracer resulted as if "different numbers of channels are activated at high and low amounts of ABA, but the behaviour of individual ion channels is sensitive to ion content (volume)" (MacRobbie, 1995, 1998). MacRobbie opted for a volume/ stretch sensitive $K^+$ channel and osmo-/stretch sensitivity was indeed demonstrated for TPK1 channels from diverse plant species (Maathuis, 2011). In contrast to FV and SV/TPC1 channels, which are equally permeable for $Rb^+$ and $K^+$ (Brüggemann et al., 1999a; Pottosin & Dobrovinskaya, 2014), VK/TPK channels hardly conduct $Rb^+$ (Gobert et al., 2007; Ward & Schroeder, 1994). Yet, a significant role of SV/TPC1 channels in ABA-induced stomata closure seems to be ruled out (Islam et al., 2010; Peiter et al., 2005), but what about FV channels? For the sake of completeness, one

needs to take also into account the coupling between the fluxes of $K^+$ and small anions ($Cl^-$), governing stomatal movements, not only for the same membrane, but also between plasma membrane and tonoplast. Indeed, plasma membrane and tonoplast communicate across the common space, cytosol. Thus, the mutation of the main anion pathway in the plasma membrane, formed by SLAC1 channel, predictably slows down the global transport across both membranes in series (Horaruang et al., 2020). Voltage dependence of the outwardly rectifying $K^+$ channel GORK in the guard cell plasma membrane senses external $K^+$, which changes manifold upon stomatal close-open transitions (Jezek & Blatt, 2017). This $K^+$ modulation ensures that at any external $K^+$ GORK channels will mediate $K^+$ efflux (Dreyer & Blatt, 2009). However, such a mechanism defines the direction of $K^+$ flux but not the condition of its cessation. A more compatible $K^+$-sensing mechanism, which is the voltage-independent control of $K^+$ channels activity by cytosolic $K^+$ was described for SKOR, another outward-rectifying $K^+$ channel, with a high degree of homology to GORK (Liu et al., 2006). But to the best of our knowledge, yet such a mechanism has not been confirmed for GORK. Thus, next we consider $K^+$ sensing mechanisms of the major tonoplast cation channels.

A focused study demonstrated that FV channels sense both vacuolar and cytosolic $K^+$ changes (Pottosin & Martínez-Estévez, 2003). An increase in the cytosolic $K^+$ resulted in a shift of the whole voltage dependence in parallel with the $E_K$ shift. In contrast, increase of vacuolar $K^+$ only affected the voltage-dependent process at cytosol negative potentials, decreasing the respective voltage threshold and increasing channel's open probability, hence potentially inducing vacuolar $K^+$ release. Overall, this analysis demonstrated that at tonoplast attainable potentials the FV activity was almost independent of cytosolic and strongly potentiated by vacuolar $K^+$ (Pottosin & Martínez-Estévez, 2003). Meanwhile, SV/TPC1 was also sensitive to vacuolar $K^+$: in the virtual absence of vacuolar divalent cations an increase of vacuolar $K^+$ resulted in the increase of the threshold for voltage activation in parallel with the right-hand $E_K$ shift (Ivashikina & Hedrich, 2005; Pottosin et al., 2005). Substitution of $K^+$ with $NMDG^+$ demonstrated that this shift was caused by shielding of the negative membrane surface charge. Yet, the effect was beyond the totally unspecific cation binding, so that vacuolar $Na^+$ caused a larger shift as compared to $K^+$ (Ivashikina & Hedrich, 2005; Pottosin & Dobrovinskaya, 2014). It was speculated then that an increase of the threshold for voltage activation by high vacuolar $Na^+$ can be a mechanism for the prevention of a $Na^+$ leak from the vacuole at salt stress (Ivashikina & Hedrich, 2005). Yet, in the presence of submillimolar $Ca^{2+}$ at the luminal side the direction of the $K^+$ ($Na^+$)-induced shift of the voltage dependence is reversed, so that the channel tends to open at more cytosol negative potentials. Luminal $Ca^{2+}$ is a principle negative regulator of the SV/TPC1 activity, and shielding the negative surface charge by $K^+$, and, more strongly, by $Na^+$, decreases the local $Ca^{2+}$ concentration in the vicinity of the membrane surface/luminal $Ca^{2+}$ binding site and alleviates the $Ca^{2+}$ inhibitory effect (Pérez et al., 2008; Pottosin et al., 2005; Pottosin & Dobrovinskaya, 2014). So that, at physiological vacuolar $Ca^{2+}$, an increased content of $Na^+$ in the lumen would *promote* vacuolar $Na^+$ release via SV/TPC1 channels. The same appears to be true for FV channels (Bonales-Alatorre et al., 2013). To summarize, at non-salinized conditions, when $K^+$ is a dominant cellular cation, both FV and SV/TPC1 channels act as vacuolar $K^+$ sensors, so that their open probability increases with the increase of vacuolar $K^+$ and decreases when vacuolar $K^+$ dropped. The latter can be suitable for a condition of extreme $K^+$ deficiency, when the electrochemical $K^+$ gradient is vacuole-directed (Walker

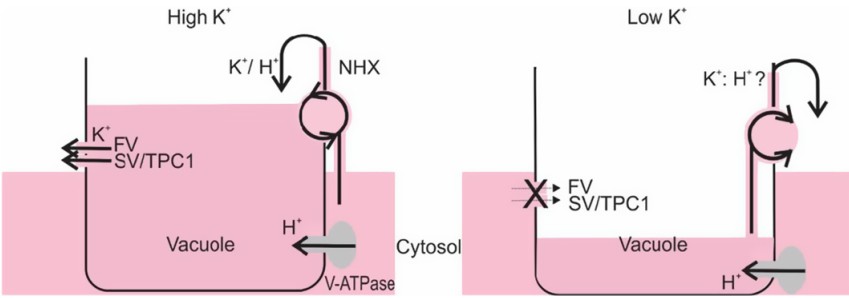

**Figure 4.** Potassium permeable channels of FV and SV types act as safety valves, preventing K⁺ overload at excessive K⁺; vacuolar K⁺ filling is mediated by K⁺ (Na⁺)/ H⁺ antiporter of NHX type (left). Suppression of the activity of FV and SV channels by low vacuolar K⁺ prevents channel-mediated vacuolar K⁺ re-uptake and allows the maintenance of cytosolic K⁺ at the expense of the vacuolar K⁺ pool. Cytosol refilling by vacuolar K⁺ can be mediated by a hypothetical K⁺ : H⁺ symporter (right). Note that in the K⁺ replete conditions a significant 'futile' H⁺ and K⁺ cycling results, whereas under K⁺ deficiency (upon the change in the direction of transtonoplast K⁺ gradient) this cycling is reduced. Above transport system acts as K⁺ sensor/effector at non-saline conditions but is unable *per se* to discriminate between K⁺ and Na⁺ under high salinity (see text).

et al., 1996), to diminish channel-mediated $K^+$ re-absorption by the vacuole (Figure 4). Yet, this sensing mechanism is not selective for $K^+$ *vs* $Na^+$, so that at salt stress additional regulatory factors need to be invoked to suppress the non-selective channels´ activity and reduce the vacuolar $Na^+$ leak (Pottosin et al., 2021; Shabala et al., 2020).

## 6. Conclusion and outlook

Potassium is the most abundant and probably also the most versatile cation in plants. Besides its rather static role as an osmotic agent and charge balancer, there is more and more evidence that it also plays significant roles in intra-plant and intra-organelle communication. In addition, potassium gradients can also serve as an important supporting energy source, for example, for phloem loading. Although our knowledge of K in plants seems to be by far the most advanced of all relevant ions, there are still many open questions that need to be answered in future research. Status of current knowledge provides the basis for a comprehensive quantitative modelling approach including properties of membrane transport proteins, volume flow and regulatory features of $K^+$. Recent computational modelling revealed the role of apparently futile $K^+$ and $H^+$ cycling for intracellular $K^+$ homeostasis and steady state membrane potential maintenance. These conclusions received an experimental confirmation, but more wet-lab work is needed to support them. The loss of $K^+$ is a common denominator of abiotic stress responses and there is an emerging view for its physiological importance as an energy switch from metabolism to defence needs. There is contrasting behaviour of different plant species and tissues in handling of intracellular $K^+$ and membrane potential in a response to stress, which raises the question of the specificity of $K^+$ sensing, decoding and downstream signalling. Some plausible $K^+$ sensing and decoding mechanisms are addressed in this review. Several $K^+$ and $K^+$-conducting channels and transporters such as SKOR, HAK5 and vacuolar SV/TPC1 and FV may be plausible candidates to act as $K^+$ sensors and effectors regulating the activity of target proteins. Therefore, the puzzle of $K^+$ homeostasis and signalling in plants is gaining shape. Obviously, to complete this job, a systemic modelling approach with focused experimentation are required.

**Data availability statement.** Data used for quantitative analysis are available on request.

**Author contributions.** S.S. conceived the concept. All authors wrote the article.

**Funding statement.** L.H.W. acknowledges a support by the National Science Foundation of China (Grant No. 32070277), I.D. was supported by the Agencia Nacional de Investigación y Desarrollo de Chile (ANID), Grant No. Anillo-ANID ATE220043 (the multidisciplinary center for biotechnology and molecular biology for climate change adaptative in forest resources; CeBioClif) and by Fondo Nacional de Desarrollo Científico, Tecnológico y de Innovación Tecnológica (FONDECYT/Chile, Grant No. 1220504).

**Competing interest.** The authors declare no competing interests.

**Open peer review.** To view the open peer review materials for this article, please visit http://doi.org/10.1017/qpb.2025.10.

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
