## [Reviewer Report]

This is a review of the roles of K+ at the subcellular, cellular, and up to the whole-plant level. It reviews some less discussed but material topics. It also introduces some hypotheses that will be useful to the field. I have some suggestions for improving the manuscript.

1. In the section on K+ signaling (pages 7 and 8), it would be good to note in the beginning (lines 242-244) that changes in K+ concentration could be a signal, as you said, but that that K+ deficiency signaling could lead to additional changes discussed here such as differences in cell fate or autophagy. K+ deficiency signaling is discussed next so it fits together.

2. In the section on long-distance transport, the role of HKT1 is not mentioned. Although HKT1 is mainly studied due to its strong effect on sodium tolerance, there are examples of HKT1 importance for potassium homeostasis. For example, by removing Na+ from the xylem, data has shown that the K+ concentration in the xylem sap is larger in wildtype controls than in hkt1 mutants in Arabidopsis, presumably due to HKT1-dependent Na+ mediated depolarisation and the ensuing K+ efflux from xylem parenchyma cells.

3. In the discussion around Fig 3, something appears to be incorrect. For Fig. 3B, line 361 “wheat K+ efflux was an order of magnitude lower and gradually increased with time”. For wheat, there appears to be a small rapid efflux that is constant over time. That would be consistent with a linear decrease in K+ concentration seen in Fig. 3C.

4. Also, Figure 3 could be improved, red and green colors are not good for two reasons, color blind readers can’t distinguish them and when printed in black and white they are not distinguishable. Change to two different shapes such as open and closed circles and squares. And some of the labeling, especially the axes and legend, appears to be low resolution.

5. The high-affinity HAK/KUP/KT K+ transporter family should be introduced and described. Only HAK5 is briefly mentioned in the Conclusions. Papers investigating the phenotypes for K+ flux and accumulation in roots of hak5 single and hak5 akt1 double mutants should be discussed.

6. lines 293 -314: With respect to the low K+ sensing and ensuing Ca2+ increase, the connection to hyperpolarisation-activated Ca2+ permeable channels should be described.

7. Studies showing effects of low K+ induced transcriptome changes should be discussed when reviewing K+ sensing.

Minor points

line 68 “with ambient K+” is not clear enough, maybe change to “with the concentration of K+ of available in the soil” or “with soil K+ ” or “with K+ in the media” depending on how the experiment was performed.

In the discussion about long-distance transport of K+ in the phloem and evidence that K+ is recycled from the leaves to the roots via the phloem, it would be useful to cite genetic evidence from Tian et al., 2010 Plant Cell. 2010 22:3963-79, who showed that when NaKR1 in Arabidopsis is knocked out, phloem is defective, and sodium and potassium accumulate in leaves. Maybe lines 91-97.

The subtitle “Cytosolic K+ levels determine cell fate” could be misinterpreted and should be more specific to the proposed hypothesis.

line 109 “SKOR is under tight control of nitrate transporter NTR1.5” it is unclear which is in control, I think this should be “”SKOR is tightly controlled by the activity of nitrate transporter NTR1.5".

line 112 “either site” should be “either side”

lines 113-116 “Hence, K+ retention at the plasma membrane, induced by an increase in volume flow passing through the cells, would favour K+ release into the xylem (so-called concentration-polarisation effect).” the meaning is not clear, maybe “Potassium uptake into xylem parenchyma cells favors K+ release into the xylem (so-called concentration-polarisation effect).”

line 221 “but only symporters and channels” not clear change to “via symporters and channels”

line 356 “withing” change to “within”

line 392 (Cuin et al. 200) should be Cuin et al., 2008.

line 439 “proved” should be “proven”, but “shown” would be better.

line 450 “slower at low ABA dose” change to “ slower at low ABA dosage”

line 467 “independent on cytosolic,” change to “independent of cytosolic K+,”

line 493 “act as safe vales” change to “act as safety valves”

line 511 “dry-lab analysis” is confusing. I believe computational modeling is meant here and would be more direct.

Line 519 the reference “Maierhofer et al” is missing.

line 522 “and hand-to-hand going focused experimentation are required.” could be stated better, maybe “ with focused experimentation are required” is better.

---

## [Reviewer Report]

Wegner et al ‘Potassium homeostasis...’

Wegner et al set out here to cover a range of topics around K homeostasis and signalling that they perceive to have been less covered in recent years. Although the abstract lists eight separate topics, the contents can be condensed to three, overarching themes, (i) long-distance K circulation, (ii) cellular K homeostasis and K sensing, and (iii) the role of tonoplast K transport in homeostasis and sensing. Some effort to condense and focus the initial descriptions might therefore be a help for the reader, as these are interwoven.

General

Overal, the issues of K circulation, and of xylem and phloem loading are explained well and the arguments for the concept of a ‘K battery’ are easily accessed. The model has been promoted by Dreyer in the past and is an intriguing one, but the presentation would benefit from some more detail to highlight the experimental evidence to support this model (l. 147-63). Equally important, readers would find useful some effort to highlight what unique predictions come from the model and how they might be tested.

Elsewhere, the presentation is strained and occasionally confused. In general, it is not obvious how the distinction is made between what determines true sensor/receptor processes and what are determined by the intrinsic kinetic features of the dynamic interactions that arise through shared intermediates, whether of the membrane voltage or of common pools of ions in each compartment. Some clarity around these distinctions is vital, if only to point out the difficulties in establishing what constitutes an ion sensory process. It is also unclear at times as to what tissues/cells the text refers to. This matter is equally important, as many of the processes are tissue-specific. Several of the detailed points below relate to these concerns.

Specific points (in order of appearance)

l.218, ff The text here and elsewhere confuses ‘steady state’ with ‘equilibria’. The two phenomena are not the same. What is described here is not a system that is ‘brought out of equilibrium’ but simply one that is displaced from its steady state. The result is a compensating change in flux that tends to return the system to the original steady state. Nothing more and no equilibrium.

Figure 2 The authors should reconsider the description of the H flux and its equilbrium here. The H-ATPase pumps one H per cycle (ATP), so the energetics imply an equilibrium voltage around -380 mV under physiological conditions, ie. pHi of 7.5 and pHo of 5.5

l.222-4 There is a long history connecting K and H flux that the authors should consider citing here. Among others, K starvation in fungi leads to a 1:1 exchange of H for K (cf. Conway & O’Malley 1944 Nature; Conway & O’Malley 1946 Biochem J) that was shown in Neurospora to arise through a H-K symport operating with the H-ATPase (cf. Rodgriguez-Navarro et al 1986 J Gen Physiol; Blatt & Slayman 1987 PNAS). The same processes and mechanisms were later shown in Arabidopsis roots (Matthuis & Sanders 1994 PNAS). There is also clear evidence in guard cells for similar mechanisms of H and K exhange (cf. Clint & Blatt 1989 Planta; Blatt & Clint 1989; Wong et al 2021 Plant Phys; to name a few).

l.227 “It former generates...” missing subject?

l. 255, ff Potassium is well-known as a factor important for many enzyme functions, if only in charge balance and protein stabilization. These are broad effects and for many, concentrations of only a few millimolar are sufficient, so well below the concentrations found in the cytosol even in K-starved cells. For the specific examples here - and generally for the benefit of readers - it would be helpful to compare and assess the concentrations of K needed for these activities and how they compare with the concentrations of K found under stress, such as evident under K starvation. If possible, it would be useful to compare the values of Ki with respect to [K] or its depletion. Otherwise, much of the proposed roles for K as a ‘switch’ are not particularly meaningful. For example, the reader must ask ‘How low must [K] go before it deactivates TORC1?’, ‘What is the Ki for TORC1 inhibition by K?’ and so on.

l. 306, ff Some clarity and detail around these experiments is called for. The Behera study cited here is easily misunderstood, principally because the experiments did not include measurements of membrane voltage. A simple explanation for the ‘K signature’ is that hyperpolarization of the membrane occurs when K is removed which leads immediately to an increase in Ca influx through hyperpolarization-activated Ca channels (and hence a rise in cytosolic Ca). Such phenomena are well-known and proven to connect through membrane voltage in many tissues (cf. Grabov & Blatt 1998 PNAS; Gelli & Blumwald 1997 J Memb Biol; Lemtiri-Chlieh & Berkowitz 2004 JBC). It is therefore unnecessary to invoke additional regulatory controls or to suggest a ‘K signature’ beyond the connection through membrane voltage.

l. 342, ff. please clarify that the reference here of voltage gating is to ion channels, not transporters in general. An additional point is missed altogether here is that coupled transporters, such as the H-Cl and H-K (HAK) symporters as well as the H-ATPase, are also under kinetic control by membrane voltage and this kinetic control is generally non-linear. The authors' own Fig. 2 makes this point clear for the H-ATPase. So even without voltage gating (of channels), membrane voltage will have very important impacts on ion homeostasis.

l.345-6 Again, the authors need to consider - and make the distinction between - regulation such as by changes in the number of active transporters (here the H-ATPase) vs the effect of voltage on the kinetics of transport. For the H-ATPase, simply depolarizing the membrane will stimulate H flux instantaneously by relaxing the voltage limitation on H-ATPase kinetics, as shown in the authors' Fig. 2.

l. 363-96 The calculations here do not take account of plasmodesmatal connections between cells and the extensive ‘buffer capacity’ that these connections confer on the root. It would be a good idea to bring considerations of cellular coupling into these calculations. Otherwise, the estimates appear wildly outside reality.

l. 367 symbols missing here

l. 402-5 The numbers here are somewhat out of date and, in all events, are highly variable. The authors will find a more recent assessment in the review of Jezek & Blatt 2017 Plant Phys.

l. 413, ff The literature around vacuolar K and Ca flux is certainly convoluted. It would be helpful for the reader if some of the details presented in this paragraph are omitted/condensed and the presentation simplified. In particular, as there is still no functional evidence for TPC1, it might be an idea to state this up front and move on.

l. 445-55 The authors would do well here to include some discussion of the OnGuard models that have proven successful in predicting the mechanics of how stomata work. The analysis that Chen et al (2012) and Jezek et al (2021) present supports the idea of the connection between content and tonoplast flux, only the mechanistic interpretation does not require stretch activated channels, just the underlying kinetic constraints of the different channels. A discussion of these connections is also to be found in Horaruang et al 2020 Plant Phys.

l. 460, ff The opening sentences are completely devoid of citation, yet the information is presented as fact. Please clarify. This entire paragraph would benefit from a rewriting. It’s almost impossible to follow, and the directionalities of the various fluxes are lost on the reader. There is much speculation also around surface charge, yet the publications cited offer little basis to make the claims here.

l.520 what is a K ‘effector’? Please clarify

Minor

l. 143 Please use English term (’Pressure-Flow') rather than the German or include it for the non-German speakers

l.174-4, 227, 416, “does not only interact”, “but hardly” and elsewhere, please consider rephrasing to avoid germanic constructions

---

## [Editor Report]

Thank you for submitting this review article to QPB. 

Copies of two independent reviews of your manuscript are attached. As you will note, the reviews are authoritative and detailed. The reviewers propose a number of suggestions that would enhance the quality of the manuscript. Major revision is recommended in the hope you will utilise the constructive comments of the reviewers to generate what I think will comprise a landmark review in the field of understanding K homeostasis in plants. 

When resubmitting the manuscript, please detail your responses to the reviewers' points.